# New Approaches to Friction Stir Welding of Aluminum Light-Alloys

**Marcello Cabibbo** [1,*], **Archimede Forcellese** [1], **Eleonora Santecchia** [1], **Chiara Paoletti** [1], **Stefano Spigarelli** [1] **and Michela Simoncini** [1,2]

[1] Department of Industrial Engineering and Mathematical Sciences (DIISM), Università Politecnica delle Marche, Via Brecce Bianche, 60131 Ancona, Italy; a.forcellese@staff.univpm.it (A.F.); e.santecchia@staff.univpm.it (E.S.); c.paoletti@pm.univpm.it (C.P.); s.spigarelli@staff.univpm.it (S.S.); m.simoncini@staff.univpm.it (M.S.)

[2] Faculty of Engineering, Università degli Studi eCampus, Via Isimbardi 10, 22060 Novedrate, Italy

[*] Correspondence: m.cabibbo@staff.univpm.it; Tel.: +39-071-2204728

**Abstract:** Friction stir welding (FSW) is the most widely used solid-state joining technique for light-weight plate and sheet products. This new joining technique is considered an energy-saving, environment friendly, and relatively versatile technology. FSW has been found to be a reliable joining technique in high-demand technology fields, such as high-strength aerospace aluminum and titanium alloys, and for other metallic alloys that are hard to weld by conventional fusion welding. Several studies accounted for the microstructural modifications induced by solid-state FSW, based on the resulting mechanical properties obtained at the FSW joints, such as tensile, bending, torsion, ductility and fatigue responses. In the last few years with the need and emerging urgency to widen the FSW application fields, broadening the possible alloy systems, and to optimize the resulting mechanical properties, this joining technique was further developed. In this respect, the present contribution focuses on two modified-FSW techniques and approaches applied to aluminum alloys plates. In a first case, an age-hardening AA6082 sheets were double side friction stir welded (DS-FSW). In a second case a non-age-hardening AA5754 sheet was FSW by an innovative approach in which welding pin was forced to slightly deviate away from the joining centreline (defined by authors as RT). In both the cases different pin heights were used, the sheets were subjected to heat treatments (peak hardening T6 for the AA6082, and annealing for the AA5754) and compared to the non-heat treated FSW conditions. Microstructural modifications were characterized by optical microscopy (OM). The mechanical properties were characterized both locally, by nanoindentation techniques, and globally, by tensile (yield, YT; ultimate, UT; and elongation, El) or forming limit curve (FLC) tests. Both the new approaches were directly compared to the conventional FSW techniques in terms of resulting microstructures and mechanical responses.

**Keywords:** FSW; aluminum alloys; mechanical properties; nanoindentation; microstructure

## 1. Introduction

### 1.1. Background

The friction stir welding (FSW) technique was developed by The Welding Institute (UK) in 1991 under the solid state welding techniques. In FSW (Figure 1) a non-consumable rotating is inserted in between the interfaces of the sheets to create inter-diffusion between the parts that constitute the joining volume of the material. The rotating pin tool produces a stirring action until the tool shoulder contacts the top surface of the sheets with a given plunge depth, thus generating a large amount of heat

by friction. As the tool moves along the welding line, the blanks are joined through a solid-state process owing to the severe plastic strain and the metal mixing across the weld. A large number of papers were published on the key features that characterize the FSW compared to other, more conventional fusion welding techniques. In these works ([1–9] to cite but few) the reader can find all the technological information concerning the major fields of application; typical components that are likely to be joined by using FSW; and the most used and appropriate alloy systems.

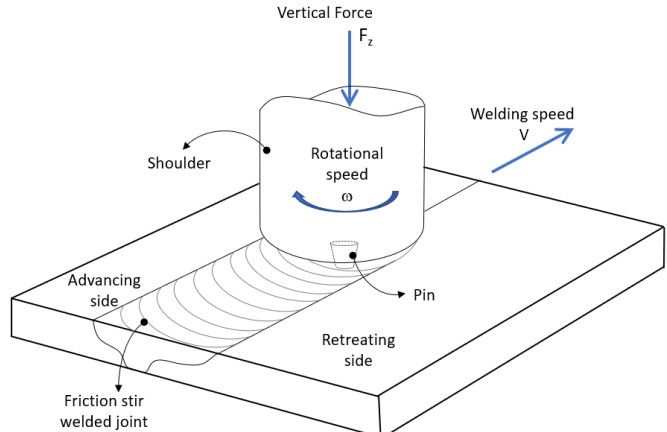

**Figure 1.** Scheme of FSW process.

One of the major advantages of using the FSW process is to overcome possible void formation possibly caused during several other fusing techniques, such as arc welding, or other clean techniques. From a technological viewpoint, FSW requires a shoulder which is protruded with a pin, also called the probe. The shoulder arrangement creates the frictional heating, and a pin probe is chiefly responsible for the generation of the material stirring throughout the joint interface volume [10]. From a microstructure viewpoint, two sides surround the joint line. An advancing side (AS) where flow of materials generally undergoes higher temperature rises, and the retreating side (RS) on the opposite side, where material flow undergoes lower temperatures [10]. The combined different temperature rises between the two adjacent sides of the joining line and the friction stirring generated by the probe, at temperatures below the material melting point ($T_M$), induce a microstructural asymmetry between the AS and RS.

It thus results that the friction welding probe (actually, in this case shoulder + pin) is one of the most important technological features involved for a successful welding process. Shoulder and pin are put in spin rotation against the fixed two parts of the joining material [11–14]. Thence, adequate knowledge of the possible optimization means to make an easy, money-saving, sound and effective weldment by FSW is nowadays a kind of mandatory goal within the scientific community active in the field of welding of light alloys. Two of the most promising optimization means for FSW refer to the metallurgical status of the joining materials, and to the FSW tool itself (chiefly the pin). Both of these issues are the object of the present work, which aims at showing the technological potentialities of making a homogeneous and highly resistant FSW joint through material thermal treatment optimization, on one hand; and on the other, at modifying the pin geometry, FSW settings and pin orientation with respect to the joining plates [15–19]. This manuscript focused on new FSW methodologies applied to light alloy plates, such as aluminum and magnesium, and to dissimilar metal plates (namely, aluminum + magnesium).

### 1.2. FSW Process Peculiarities

FSW is known as a relatively easy and reasonably cost-effective joining techniques, as it can be used with a conventional vertical milling machine. The technique uses virtually non-consumable and economical tools (both shoulder and pin), and no filler material or shielding gas is required to join the material. FSW is chiefly used to join light alloy plates and sheets, and thus a relatively low

energy is required to weld, and no complex or costly fixture devices are generally required for holding the work piece. Here are some of the input process parameters which affect the output properties of the final weld product: welding speed (v); rotational speed (ω); tool geometry and shape; blank thickness; energy input; pressure applied; tilt angle; specimen grooving; tool plunge length; dwelling time; sheet orientation with respect rolling direction; the blank's material; the metallurgical status of the joining plates (mainly the pre-/post-heat treatments). Those are some of the FSW input variables upon which micro and macromechanical properties of joints depend. In particular, it was demonstrated that, among process parameters, the rotational speed and welding speed have a strong effect on heat generation, heat dissipation and cooling rate. Thence, joint formation, microstructure of joints and forces developed during FSW process are significantly influenced by ω and v values [1,3,7,15,17,20–30].

As for the resulting plate microstructure modification at the welding zone and along the adjacent zones, three different metallurgical zones are usually recognized to which peculiar grained structure, secondary phase precipitation sequences and mechanical properties correspond.

These are (Figure 2):

a. Nugget zone (NZ): This is where the metal is in direct contact with the pin being continuously stirred during the passage of the rotating pin, thus creating the necessary strong bond between the two metals under the welding. Fast thermomechanical heating and cooling occur, and they favour the occurrence of recrystallization phenomena. Thence, fine grain structures in the form of onion rings are generally formed.

b. Thermomechanically affected zone (TMAZ): Where the microstructure is highly plastically deformed. In this zone the microstructure experiences a significant grain morphology and size modification with occurrence of secondary phase particles precipitation or coarsening, in the cases of age-hardening aluminum and magnesium alloys. Although the TMAZ undergoes plastic deformation, recrystallization does not occur in this zone due to insufficient deformation strain.

c. Heat affected zone (HAZ): Where the material undergoes thermal cycles with no plastic deformation. In this zone the microstructure is characterized by retaining same grain structures as the parent material.

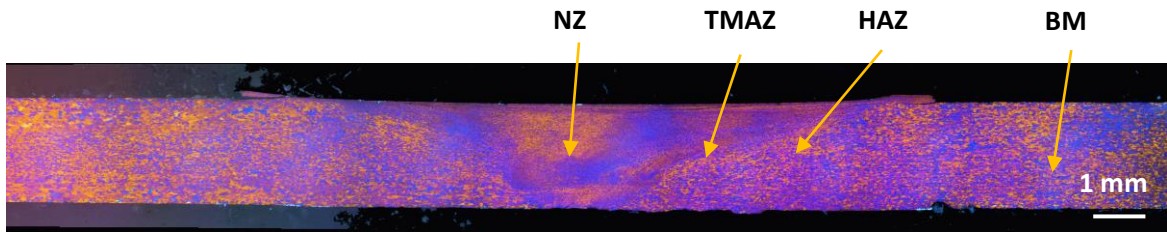

**Figure 2.** Typical optical microscopy macrograph showing various microstructural zones in a FSW joint in AA6082-T6 aluminum alloy (blanks thickness = 2 mm, truncated cone pin tool with shoulder diameter = 12 mm, ω = 1200 rpm and v = 100 mm/min).

The NZ is the welded plate zone typically as wide as the shoulder size, the TMAZ is the immediately surrounding welded affected zone whose extension depends on several pre-set parameters, such as shoulder transverse speed, pin geometry and orientation with respect to the joining plates, the plate material and the material metallurgical status prior to welding. Furthermore, the HAZ depends on the above mentioned FSW settings, but its extension is usually much less than the TMAZ one. HAZ is recognized as it is characterized by having a grained microstructure similar to the BM microstructure and significantly different from the TMAZ microstructure, which deeply differentiates from the other two by the presence of the plastically deformed grained structure [1,16,26].

The base metal (BM) generally consists of elongated grains oriented along the drawing direction. At this distance the BM is unaffected by the thermal and microstructure modification induced by the

FSW. In other worlds, the BM is defined as the minimal lateral plate distance where the microstructure does not experience any effect of welding.

### 1.3. Outlines of FSW Applied to Aluminum Alloys

Several different settings and experimental conditions have been tested in light alloys so far. Some of the most significant, promising and technologically sound are hereafter briefly reported.

FSW tools consisting of columnar probes with threads and without threads, and a triangular prism, were used by Fuji et al. [31] for butt welding of 1050-H24, 5083-O and 6061-T6 aluminum alloys. They found that whenever metal deformation resistance was lower, a columnar tool without threads was the effective setting solution, whereas in the presence of more resistant alloys, a columnar tool without threads was shown to be a proper solution only by decreasing the rotation speed.

Different welding pin rotations and transverse speed, and pin pressures, were tested by Wei et al. [32] on Al-Li alloy plates. They reported optimal rotational and transverse speed to obtain sound and grooveless welded plates, as four inappropriate speed tunnel type defects were observed. On the other hand, pin axial pressures which were too high were reported to generate material depression and wavy burrs across the NZ, while for too low an axial pressure tunnel, a groove type defect occurred.

Additionally, Patil et al. [33] investigated the effects of different welding transverse speeds, with constant pin rotational speed and pin profiles, on the weld quality of AA6082-O plates. In their study it was reported that joint fabricated by using taper screw thread pin exhibited superior tensile strength to using a tri-flute pin, irrespective of welding speed. Moreover, a maximum hardness reduction was obtained by using the threaded, conical and trapezoidal pin profiles, while the use of tapered, cylindrical and trapezoidal pin geometries generated a significant ductility increase in both NZ and TMAZ/HAZ compared to that of the BM.

The combined effect of pin axial pressure and pin profile types on the meaningful FSW zones were studied by Elangovan et al. [34] in AA6061 aluminum alloy plates. In their study a clear relationship between pin geometry and optimal pin pressure was found. In particular, straight and tapered cylindrical pin profile needed an optimal pin pressure of some 6 kN (value depending on the material and plate thickness). On the other hand, by using threaded cylindrical pin profile and square pin and triangular pin profiles, sound, defect-free weldments were reported irrespective of the pin axial pressure. In this respect, joints produced using square and triangular pin profiles showed superior tensile properties compared to the use of a straight cylindrical pin profile.

The role of material heat treatments on the FSW joint quality and optimization of aluminum alloys was studied by Zhang et al. [35]. In particular, they addressed the HAZ as it is generally the intrinsically weakest location of a normal, friction stir welded, precipitate hardened aluminum alloy series. A proper cooling, and possibly, low-temperature friction processes, were reported to meet an optimization of the FSW and high tensile strength of the age-hardened aluminum alloy welded plates.

As for dissimilar alloy plate welded by FSW, several studies reported specific FSW settings to improve the joint quality. These were also coupled with specific metallurgical conditions of the starting plates, and also of the post FSW dissimilar alloy plates. Some metallurgical problems are likely to be involved in the FSW of two aluminum alloy plates where only one is an age-hardened alloy series. In this respect, some peculiar metallurgical problems that deserve to be here mentioned were reported by Koilraj et al. [36], who used an Al-Cu/Al-Mg plate system. Indeed, it is known that fusion welding of dissimilar aluminum alloys is quite challenging due to the formation of low melting eutectic by the constituent element, resulting in cracking. Thus, a significant hardness reduction on the age-hardening alloy plate from the BM throughout the NZ is likely to occur, while on the contrary, this usually does not occur on the not-age-hardened alloy plate. The reported hardness reduction is generally higher in the advancing side (AS) than the one occurring in the retreating side (RS). To overcome this metallurgical asymmetry and inhomogeneity, several approaches can be taken; these include post-welding joint plate homogenization, double side friction stir welding (DS-FSW), small pin

deviation from the welding line during the pin translation motion and some others concerning shoulder diameter and pin geometry/orientation to the joining plates. In this respect, the first two techniques are herein presented and discussed as the most promising solution to overcome the microstructural and mechanical inhomogeneities between the AS and the RS.

Another important welding parameter that has been outlined by Tutar et al. [37] in a study on non-heat treatable AA3003-H12 refers to the pin depth into the plate thickness. In fact, they reported a noticeable tensile strength increment with pin depth into the plate thickness, and accordingly all the meaningful FSW zones, NZ (stirred zone, SR), TMAZ and HAZ, expanded.

AS for modelling of the effect of different FSW parameters and setups, a study by Kumar et al. [38] investigated the effects of pin rotational and transverse speeds, and axial load, on the tensile strength and ductility of dissimilar AA5083-O and 6061-T6 plates. These mathematical models showed increased tensile strength with both axial load and pin speed incremented up to a certain maximum level, at which the material strength started to decrease. This is due to an insufficient stirring at the highest welding speeds, and then the material in the AS does not travel enough to the RS, causing the material defects and local sites of incomplete welding. On the other hand, as pin rotation and axial force rise, the ductility increases accordingly, whereas as pin transverse speed rises, the ductility decreases. Accordingly, Kadaganchi et al. [39] formulated a mathematical model with spindle speed, welding speed, tilt angle and tool geometry to predict the yield strength, tensile strength and ductility of FSW AA2014-T6 plates.

### 1.4. Typical Microstructure of FSW Joints

The characteristic microstructural zones of a typical FSW joint are deeply driven by the thermomechanical cycles to which the joining plates are subjected during welding. In this sense, transients and gradients in strain, strain rate and temperature are inherent in the thermomechanical cycles of FSW. Material heating is generated by a combination of pin and shoulder friction, and by severe, localized, adiabatic deformation induced in the material by the pin rotation. Thus, the heat source is distributed in a volume of deforming material surrounding the pin. As already mentioned, the material directly in contact with pin and shoulder is referred to as the weld nugget zone (NZ), or stirred zone (SZ). During FSW, material flows in a complex, vortex-like pattern around the pin from the AS (defined as tangential velocity of a point on the pin surface parallel to the traversing direction) to the RS (defined as tangential velocity of a point on the pin surface antiparallel to the traversing direction). By controlling the pin rotation and transverse speeds, the thermomechanical cycles undergone by the different meaningful FSW zones can be predicted and optimized.

The NZ is strongly affected by the pin geometry and rotation speed, and it generally experiences large plastic strains [40]. Peak temperatures reached in the NZ can range 0.6 to $0.95T_M$, depending on the material, pin geometry, and other operating conditions. On its upper surface, extra heating and deformation also occur by the effect of the shoulder attrition [41,42]. From a microstructural viewpoint, the NZ is generally characterized by a fine, or even very-fine equiaxed grained structure. In fact, NZ refined grained structure can even reach a nano-scale size (< 100 nm), as reported in [43] for an AA7000 FSW plate.

Immediately next to the NZ is the TMAZ where the material experiences lesser strains, strain rates and lower peak temperatures. This region is often characterized by a pattern of grain distortion that suggests shearing and flow of material about the rotating pin. The grain distortion may lead to fragmentation and formation of refined, equiaxed grains near the TMAZ/NZ interface, which is actually a progressive microstructure transformation inter-zone.

Beyond the TMAZ is the HAZ, which does not experience any strain deformation, but only a thermal cycle, with temperature peaks as low as typically one-fifth of that of the NZ.

The FSW induced microstructure modifications of chiefly the NZ and the TMAZ, were also modelled by classical dynamic recovery (DRV) and dynamic recrystallization (DRX) phenomena induced by the thermomechanical stirring during FSW [44,45].

As for DRV, this occurs during hot working of high stacking fault energy (SFE) metallic materials, such as aluminum [46–48]. As the rate of DRV increases, the dislocation density increases, and most of them start to rearrange to form low-angle boundaries (LABs). Eventually, the flow stress saturates, as hardening due to dislocation multiplication and recovery due to dislocation rearrangement reach a dynamic equilibrium. This leads to deformation at a steady state wherein the flow stress remains constant with strain. The steady state is reflected in equiaxed subgrains (LABs) with nearly dislocation-free and even grains (high-angle boundaries, HABs). Typical examples of the different microstructures formed during FSW (as well by FSP) can be found in published works, such as [16,17,49] to cite but few.

As for continuous or discontinuous DRX, this occurs in medium to low SFE metallic materials during hot deformation [46]. In DRX, new, dislocation-free grains form at sites such as prior grain boundaries, deformation band interfaces or boundaries of newly recrystallized grains. During deformation, a low stacking fault energy retards the climb and cross-slip processes of DRV, thereby enabling the formation of stable grain nuclei at various sites in the deformation microstructure. In other words, the microstructure, which forms under and actually forms after DRX, is characterized by almost all grain boundaries with an irrelevant presence of tangled dislocations inside. This microstructural aspect marks the distinctive difference between the NZ and the adjacent TMAZ. Both these thermomechanical deformation mechanisms occur as the deformation temperatures exceed $0.3T_M$, which is always the case of the TMAZ and NZ of FSW light-alloy joints [50–52].

All the herein mentioned mechanisms of formation for sub-grains and grains (TMAZ) and recrystallized fine-grains (NZ) are always also dependent on the material initial metallurgical conditions and were subjected to different processing and tool parameters (both pin geometry and orientation to the plate, and shoulder size).

A further argument that is likely to have to be taken into account when the different FSW zones are studied and characterized consists of the transients and steep gradients that are generally generated by the passage of the pin during FSW.

In the present contribution, in order to optimize both microstructural modifications and mechanical responses of light-alloy welds, two new techniques based on FSW technology are presented. The first such new technique is a double-side FSW (DS-FSW) applied to an age-hardened AA6082 alloy; the second consists of a little deviation from the pin transverse welding line during FSW applied to aluminum alloy plates. In both cases, the role of the material metallurgical status prior to and after FSW is addressed, and the discussion accounts for the improved results from both a microstructural and mechanical viewpoint.

## 2. New Approaches to the FSW Joining Techniques for Light Alloys

### 2.1. Experimental Setup

FSW joining was carried out by a computer numerical control (CNC) machining centre to obtain butt joints on sheet blanks. Probes with conical pins of H13 steel (HRC = 52) of different pin heights were used. The probes had a shoulder diameter of 15 mm, a pin base width of 3.9 mm and a 30° pin angle with respect to the shoulder surface. To obtain different tool sinking depths, two pin heights of 2.0 and 2.3 mm were used. The welding line was perpendicular to the aluminum alloy rolling direction. A vertical force of 1.7 kN was used. FSW rotational speed, $\omega$, and traverse velocity, $v$, were imposed equal to 1200 rpm and 100 mm/min, respectively. All the reported experimental tests, both microstructural and mechanical, result from three individual measurements. Thence, the results shown represent mean behaviour among the three samples.

### 2.2. Double Side Friction Stir Welding Methodology

FSW usually induces localized high stress levels that in some cases can lead to joint failures [53], mainly caused by the non-uniformity of grain size at the SZ; formation of microcracks and voids; and

poor formability respect to the base material (BM). A further drawback potentially related to FSW of age-hardening aluminum alloys is the non-uniformity of both precipitation and grain size at the NZ. This indeed, is attributed to temperature inhomogeneities across the NZ, and to different strain and strain to which both NZ and TMAZ are subjected during the welding process [7,54–56]. Therefore, there is a strong need for an improvement in ductility and formability of FSW aluminum joints. Studies reported significant mechanical improvements by using multi-pass, double lap, reverse dual-rotation FSW and FS spot welding ([16] and references therein). Anyhow, by using the above-mentioned improved techniques, neither a fully satisfactory microstructural uniformity across the NZ, nor a formability response from these FSW joints, was reached. In this respect, welding on both sheet surfaces is believed to be able to yield a better response in the strength, elongation and the formability of the FSW joints. This new approach is identified here as double side friction stir welding (DS-FSW), and it is intended to induce the serration of the geometric discontinuities and to induce a significant microstructural homogeneity at the NZ. Moreover, DS-FSW was developed in order to overcome typical drawbacks generated by FSW process, such as the limited post-welding formability of FSW joints due to both the thickness reduction in the weld resulting from the forging effect of the shoulder, and the small geometric discontinuity into the joint, located at the bottom surface of the sheets, that causes local increase in stress and acts as a notch [57–59].

The DS-FSW methodology involves performing the FSW process on both blanks surfaces: the first welding operation is followed by a second one carried out by placing the rotating tool in contact with the sheet surface opposite to the one welded by the first pass (Figure 3). DS-FSW was successfully applied in joining blanks in AA6082-T6 aluminum alloy, using a pin tool (Figure 4a) for the first welding pass and a pinless tool (Figure 4b) for the second one [16,17]. The material investigated was AA6082 aluminum alloy, in the T6 temper state, supplied in the form of 2-mm thick sheets. The blanks, 180 mm in length and 85 mm in width, were cut in order to perform FSW operations with the welding line perpendicular to the rolling direction. Welding processes were performed using constant values of the rotational speed ($\omega$) and welding speed (v) equal to 1200 rpm and 100 mm/min, respectively. A nuting angle of 2° was imposed during welding operations.

The post-welding formability exhibited by DS-FSW joints was analysed by performing hemispherical punch tests (Hysitron$^{TM}$ Inc., Minneapolis, MN, USA) according to International Standard ISO 12004 [60]. The equipment used consisted of a die, a blank holder and a hemispherical punch with a diameter equal to 30 mm. The tests were carried out with a punch speed of 0.1 mm/s. Post-welding formability was evaluated by means of both limiting dome height (LDH) and forming limit curves (FLCs) at the onset of necking. Toward that purpose, samples with width to length ratios equal to 1 (100 mm × 100 mm) were clamped and deformed until necking occurred, in order to measure the LDH value, representing the punch stroke at the peak value of the punch load vs. punch stroke curves. As far as the forming limit curves are concerned, samples characterised by width to length ratios ranging from 0.125 (12.5 mm × 100 mm) to 1 (100 mm × 100 mm) were used, with the length side being parallel to the welding line positioned in the central zone of the sample, as shown in Figure 5. More details on equipment and sample arrangements are available in [17]. In order to evaluate the strain distribution after hemispherical punch test, the top surfaces of samples were meshed using a regular line grid with 1.5 mm in line distance. An accurate image analysis system was used to measure the major ($\varepsilon_1$) and minor ($\varepsilon_2$) strains. The microstructural modifications and the local mechanical response, namely, hardness and elastic modulus, were also investigated. To those ends, light-optical overviews of the FSW blanks were performed using a Reichert-Jung™MeF-3$^{®}$ (Electronic Instruments Group, EIG of Ametek, Inc., Berwyn, PA, USA) microscope. Finally, nanoindentation was used to mechanically characterize the different welded zones. The indentation measurements were carried out using a Hysitron$^{©}$ UBI$^{®}$-I (North Carolina State University Raleigh, Raleigh, NC, USA) nanoindenter, equipped with a motorized stage. A Berkovich diamond tip was utilized. The tip was calibrated with a fused quartz reference sample. More details are available in [16]. Tests under same conditions were repeated at least three times for assuring repeatability of results.

In order to evaluate the advantages offered by the new welding methodology, the experimental results obtained using the DS-FSW were compared with those given by the conventional FSW, carried out, under the same process conditions, both in the pin and pinless tool configurations.

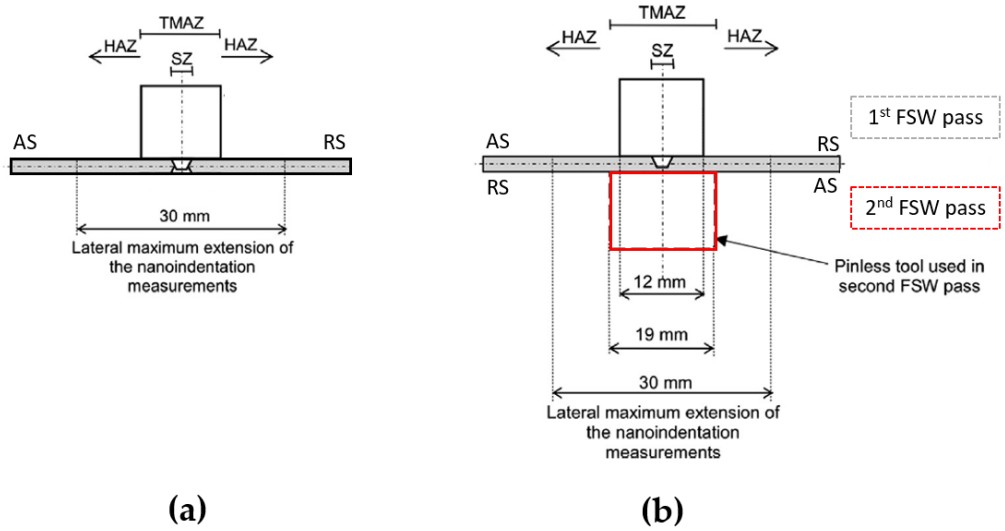

**Figure 3.** Comparison between (**a**) conventional FSW and (**b**) DS-FSW processes.

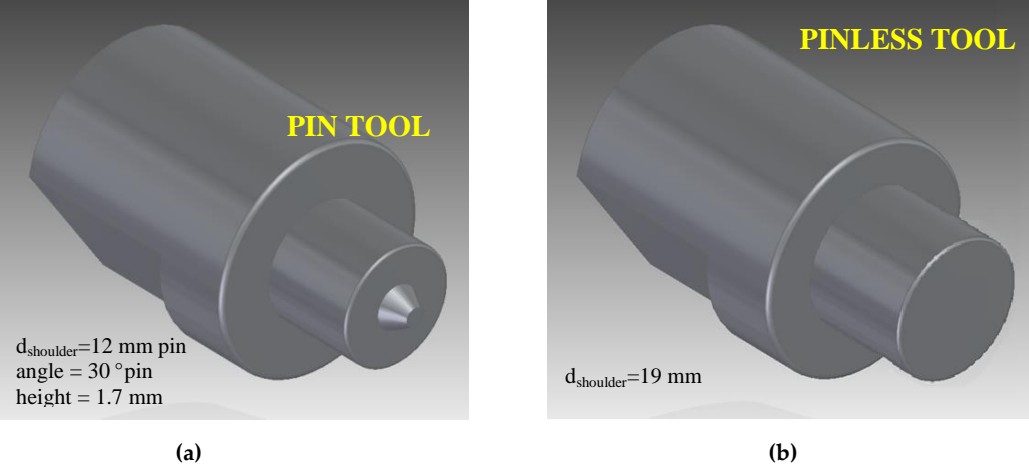

**Figure 4.** (**a**) Pin and (**b**) pinless tools used for DS-FSW process of 2-mm thick blanks in AA6082-T6 alloy.

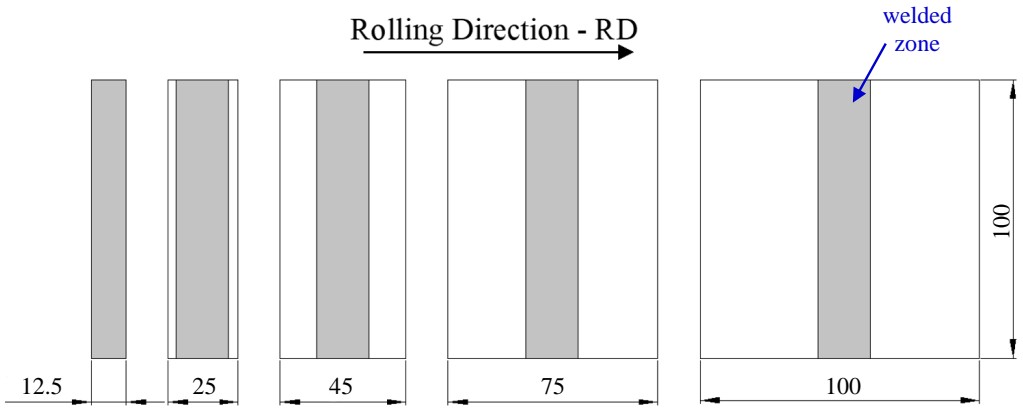

**Figure 5.** Welded samples used for the hemispherical punch test (data are reported in mm).

### 2.3. RT-Type Friction Stir Welding Approach

During FSW, an oxide layer on the butt surface is known to be likely to form. A serious defect associated with the oxide layer is the "kissing-bond" phenomenon; that is a partial remnant of the un-welded butt surface below the NZ attributed to insufficient plunging of the welding tool during FSW. Formation of the kissing-bond is related to the insufficient break-up of oxide layer by an insufficient stretch of the contacting surfaces around the welding probe. This in turns can be formed by low heat-input parameters [18,20]. Moreover, FSW induced stress and strain rate, which are primarily generated by the pin and the shoulder motions across the sheet/plate, are the main technological terms for the design optimization of pin and shoulder. These, in turn, also determine the durability of the final joined material [61,62]. In this respect, some recent works reported on good-quality welds obtained by using stationary shoulder tools where the rotating pin generates the friction heat necessary to effectively join the abutting surfaces [63]. Anyhow, works in the literature have cast sufficient light on the material flow during FSW of similar [64–71] and dissimilar metallic materials [65,72–74]. Thus, different methodologies were used to analyse the flow mechanisms and to address a variety of welding conditions and set-ups; i.e., different metals, pin geometries and process parameters [75].

In this framework, Cabibbo et al. developed an innovative approach to the conventional FSW process, defined by authors as RT-type configuration, in which the welding motion of the pin tool was obtained by combining two different movements occurring simultaneously [18]: (i) the rotation of the pin axis around an axis, perpendicular to the sheet blanks and belonging to the welding line, with a radius equal to R, and (ii) the translation of the pin axis along a direction parallel to the welding line (Figure 6). Two different R values, equal to 0.5 and 1 mm, were considered. For this study, non heat-treatable AA5754 sheets 2.5 mm in thickness were used. The influence of annealing, prior and after FSW, was also investigated. The new RT-type methodology was compared with the conventional one, in which the welding motion occurs linearly along the welding line (i.e., R = 0). Both RT-type and conventional FSW processes were carried out with a rotational and welding speed equal to 2000 rpm and 30 mm/min, respectively.

The effects of the radius R and annealing treatment on microstructural, and micro and macromechanical properties was studied in order to define the process conditions and the temper state that allows one to obtain defect-free joints, avoiding the typical oxide defects of kissing-bonds, and the faint zigzag line pattern in the NZ.

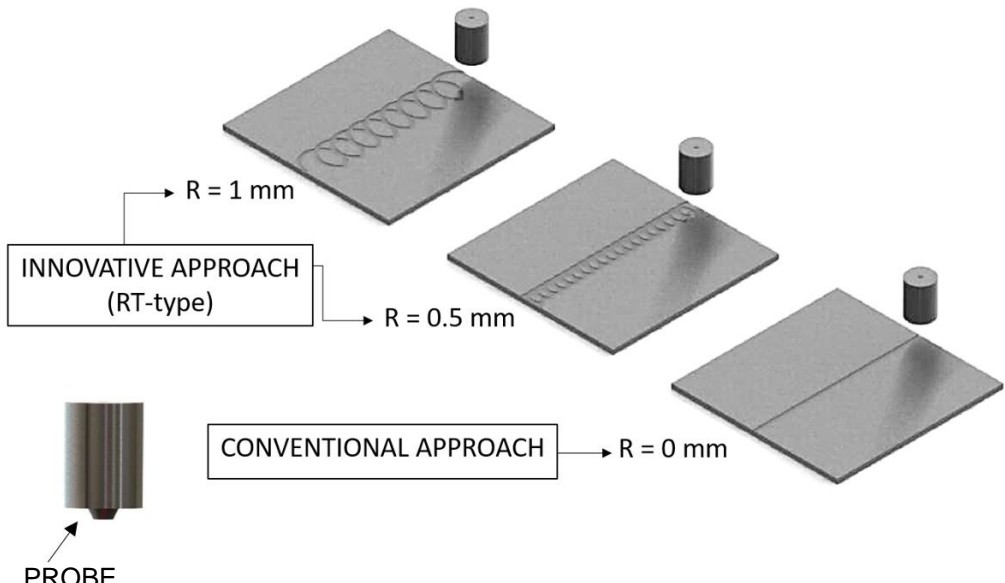

**Figure 6.** Tool photography and schematic view of the different FSW approaches used.

## 3. Results and Discussion

According to [7], the friction stir welding process is mostly used to join "unweldable" materials, such as aluminum alloys, since they do not reach melting temperature and solidify during the welding process. Moreover, FSW is receiving increasing interest owing to the energy efficiency, environment friendliness and versatility that make the process a promisingly ecologic and "green" technology [22,25]. Furthermore, the mechanical properties and post-welding formability of joints in aluminum alloys manufactured by FSW are known to be slightly higher respect to laser welding.

In order to improve the mechanical properties and post-welding formability of friction stir welded joints of aluminum alloys, Cabibbo et al. proposed two innovative approaches to the FSW technology, the DS-FSW and RT-type FSW [16–18].

Table 1 reports the limiting dome height values of the welded joints, both for the conventional FSW and DS-FSW processes. Irrespective of the welding methodology used, LDH values of joints are lower than those measured for the base material, due to the presence of the welding line that leads to a reduction in formability [60,61,74–78]. By focusing on the DS-FSW, it can be observed that the joints are characterized by LDH values higher than those measured on conventional FSW joints, leading to a reduction of LDH, as compared to the base metal, of 10.9%; compare that to 27.7% between the conventional joint FSW and the BM [17].

**Table 1.** Limiting dome height (LDH) values obtained by hemispherical punch tests on base material and joints obtained by conventional FSW and DS-FSW methodologies.

| Plate Condition | LDH (mm) | Reduction in LDH with Respect BM (%) |
|---|---|---|
| BM | 11.9 | - |
| FSW | 8.6 | 27.73 |
| DS-FSW | 10.6 | 10.92 |

The improvement given by DS-FSW can be attributed to the positive effect of the second pass that allows both the closure of the geometric discontinuity and the decrease in the height of the step generated by the first welding pass. Such results can be attributed to the different microstructures obtained by the two welding methodologies [16]. In particular, DS-FSW joints (Figure 7a,b) are characterized by a noticeable recrystallized grained homogeneity across the stirred zone, whilst, in the conventional FSW (Figure 7c,d), coarser recrystallized grains can be observed near the top surface of the joint, as compared to the grain size at the opposite bottom surface. Moreover, as far as the DS-FSW joints are concerned, the hardness (Figure 8a) and the elastic modulus (Figure 8b) measured along different profiles of the joint section are characterized by a more uniform behaviour with respect to the conventional FSW joint, in which needle-like $(Fe,Mn)_3SiAl_{12}$ intermetallic particles appear (Figure 7c,d). In the conventional FSW, low hardness and Young's modulus values are obtained in the retreating TMAZ (Figure 9); in particular, these values accounted for a drastic hardness reduction of about three times, and an elastic modulus reduction of almost ten times, with respect to the those provided by base material [16].

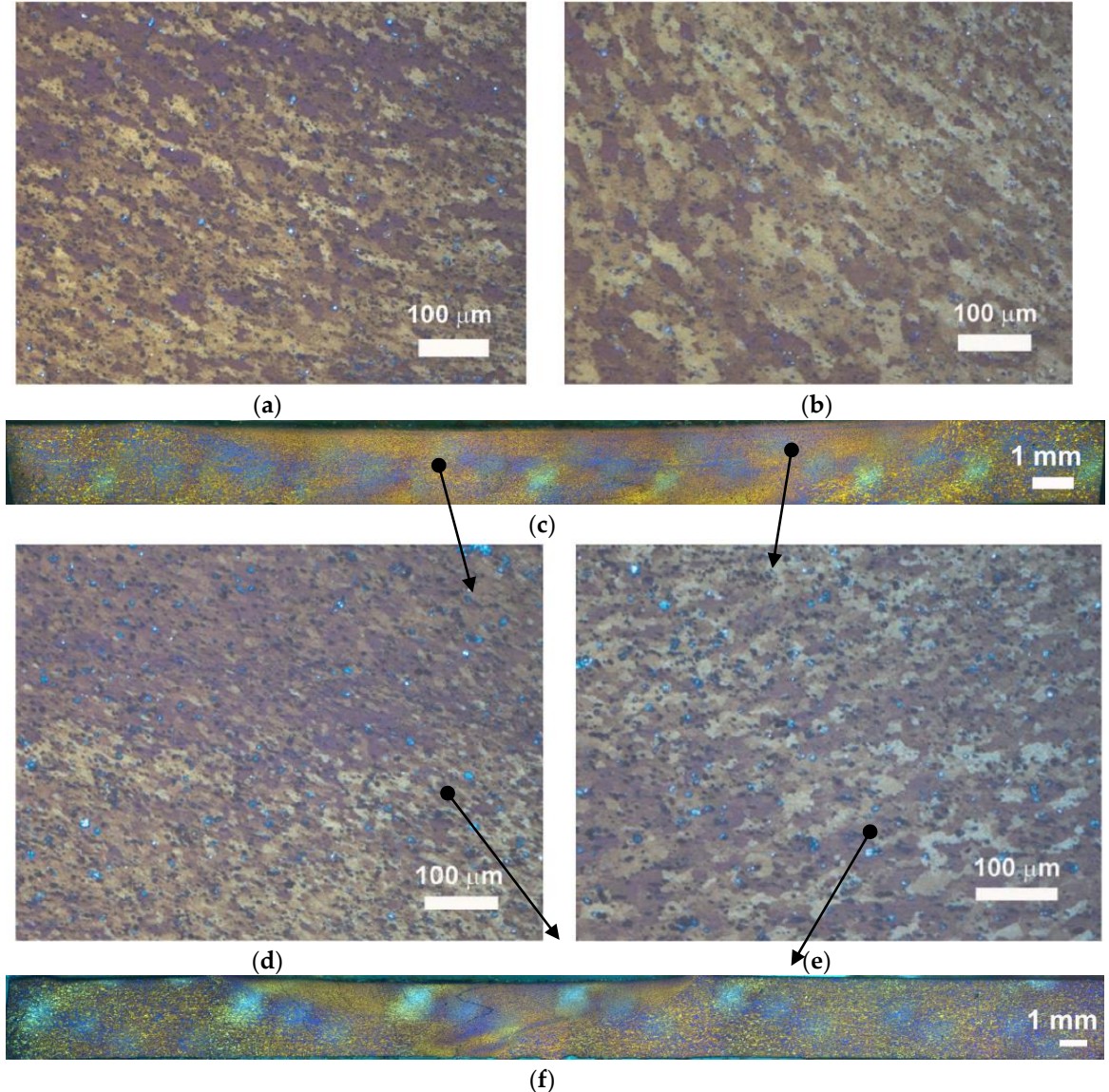

**Figure 7.** Polarized optical micrographs (POM) of the microstructure in the different zones of the welded joints in AA6082 obtained under different process methodology: (**a**) nugget zone (NZ) and (**b**) thermomechanically affected zone (TMAZ) of DS-FSW joints obtained using pin–pinless configuration; (**c**) a POM montage showing the whole DS-FSW section from which the detailed POM (**a**) and (**b**) were taken; (d) NZ and (e) TMAZ of a conventional FSW joint using pin configuration; (**f**) a POM montage showing the whole conventional FSW section from which the detailed POM (**d**) and (**e**) were taken. (v = 1200 r/min; v = 100 mm/min)

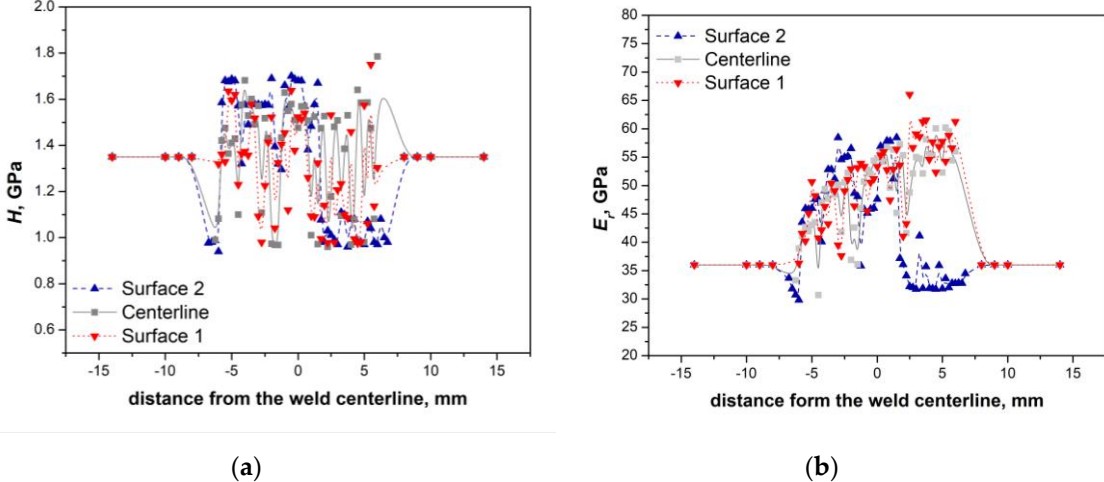

(**a**)  (**b**)

**Figure 8.** (**a**) Nanoindentation hardness and (**b**) nanoindentation Young's modulus along different profiles of the cross-section of DS-FSW joints (Surface 1: 10 μm from the top surface of joint; centreline: at a middle height of thickness sample; Surface 2: at 10 μm from the bottom surface of joint).

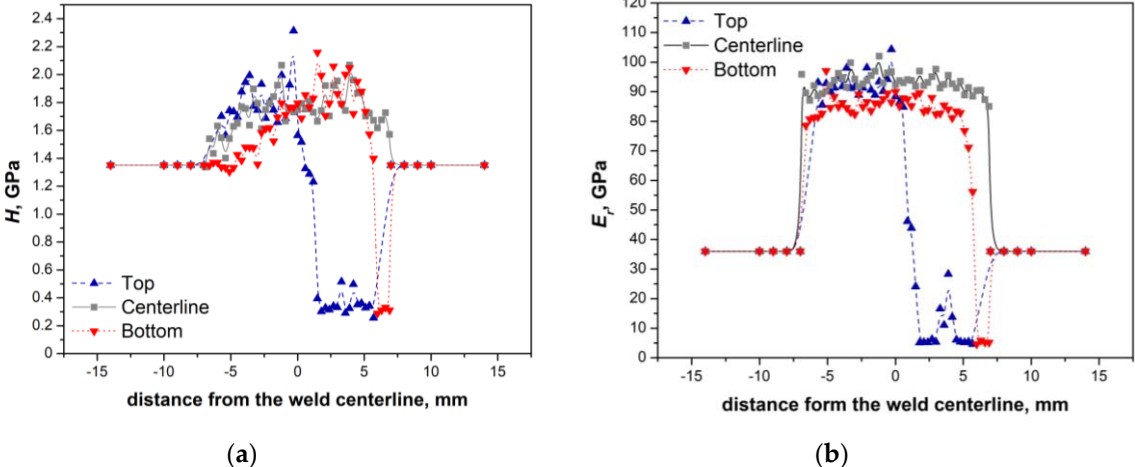

(**a**)  (**b**)

**Figure 9.** (**a**) Nanoindentation hardness and (**b**) nanoindentation Young's modulus along different profiles of the cross-section of conventional FSW joints (Surface 1: 10 μm from the top surface of joint; centreline: at a middle height of thickness sample; Surface 2: at 10 μm from the bottom surface of joint).

Figure 10 compares the forming limit curves obtained by DS-FSW joint and by a conventional FSW joint. In particular, according to the results shown in Table 1 in terms of the limiting dome height, it can be observed that, for a given minor strain value, the major strain provided by the DS-FSW joint is systematically higher than that of a conventional FSW joint. The higher vertical position of the FLC obtained by the DS-FSW sample than that of the conventional FSW ones confirms the improved formability of the DS-FSW joints due to the more homogeneous recrystallized grained structure, the local elastic modulus uniformity across the weld and the less dramatic hardness variation in the DS-FSW welds.

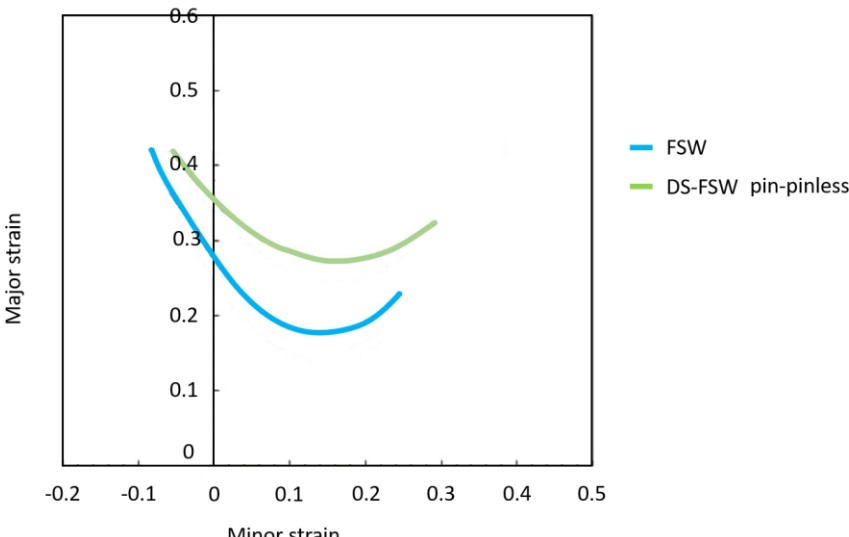

**Figure 10.** The effect of welding methodology process, and a sample arrangement of welded joints versus base material on forming limit curves.

In order to overcome typical FSW defects in 5000 aluminum series alloy joints, such as the oxide layer formation on the butt surface leading to the "kissing-bond" phenomenon and the faint zigzag-line pattern, the RT-type FSW methodology leads were proposed by Cabibbo et al. [18]. The formation of such defects, depending on the tool geometry and shape, and process parameters, significantly affects the mechanical properties of joints [15,77,78].

Table 2 reports the mechanical properties, in terms of ultimate tensile strength (UTS) and ultimate elongation in percentage (UE), of joints obtained by FSW both using the conventional approach and innovative RT-type configuration one. The effect of the annealing temper state after the FSW process on the UTS and UE values can also be observed.

**Table 2.** Comparison between the mechanical parameters of RT-type FSW joints and conventional FSW ones evaluated by uniaxial tensile tests.

| Mechanical Properties | Un-Welded Material | | Conventional FSW R = 0 mm | | Innovative RT-Type FSW | | | |
|---|---|---|---|---|---|---|---|---|
| | | | | | R = 0.5 mm | | R = 1 mm | |
| | as Received | Annealed | as Received | PWA | as Received | PWA | as Received | PWA |
| UTS (MPa) | 260 ± 10 | 230 ± 10 | 230 ± 10 | 130 ± 10 | 190 ± 10 | 240 ± 10 | defects | 220 ± 10 |
| UE (%) | 23 ± 1 | 37 ± 1 | 18 ± 1 | 3 ± 0.5 | 5 ± 0.5 | 26 ± 0.5 | defects | 18 ± 1 |

By considering the blank materials in the as received condition, the mechanical properties of joints are significantly influenced by the R value; in particular, the UTS and UE values decrease as R value increases from 0 up to 0.5 mm, whilst for R = 1 mm, no sound welds were obtained. The different mechanical properties of joints can be attributed to the different microstructure induced by the FSW process, as can be observed in Figure 11. In particular, a sound joint, characterised by a consistent stirring of the grained structure in the NZ, without void formation or de-bonding defects, is obtained by performing conventional FSW; i.e., R = 0 mm (Figure 11a). A sound joint is also provided by the RT-type FSW performed with R = 0.5 mm; in this configuration, the grain stirring in the NZ appears more pronounced than that exhibited by joints obtained by conventional FSW (Figure 11b). Pronounced defects, such as microvoids; microcracking; Al2O3 oxides; and some traces of surface de-lamination and de-cohesion, characterize the joint given by the RT-type FSW process with R = 1 mm. Mechanical properties and microstructures of the welded joints, from welding blanks of AA5754 alloy

in the as received condition, showed that FSW in the RT-type configuration induced a reduction in the mechanical response of joints as compared to the conventional FSW process.

A marked improvement in micro and macromechanical properties of joints was obtained by performing a post-welding annealing treatment (PWA) at 415 °C/3 h, followed by furnace cooling. As a matter of fact, Table 2 shows that UTS and UE values improve as PWA is performed on RT-type FSW joints. Moreover, it can be seen that FSW joints obtained by welding with R = 0.5 mm and then subjected to annealing treatment are characterised by mechanical properties similar to those given by the base material in the annealed temper state: the UTS was 15% higher, with a ductility reduction of within 30%, as compared to the un-welded annealed blanks. In such conditions, the microstructure of the NZ appeared to be decorated by very coarse grains, generated by a geometric, dynamic recrystallization mechanism produced by the combined effect of the shoulder pressure (heat input) and the post-welding annealing thermal energy. On the contrary, the behaviour exhibited by conventional FSW joints and then treated by annealing is very different from the one provided by RT-type FSW joints since both UTS and UE values drastically decreased compared to the annealed base material. In such an approach, traces of microstructural debonding near the centre of the NZ appear.

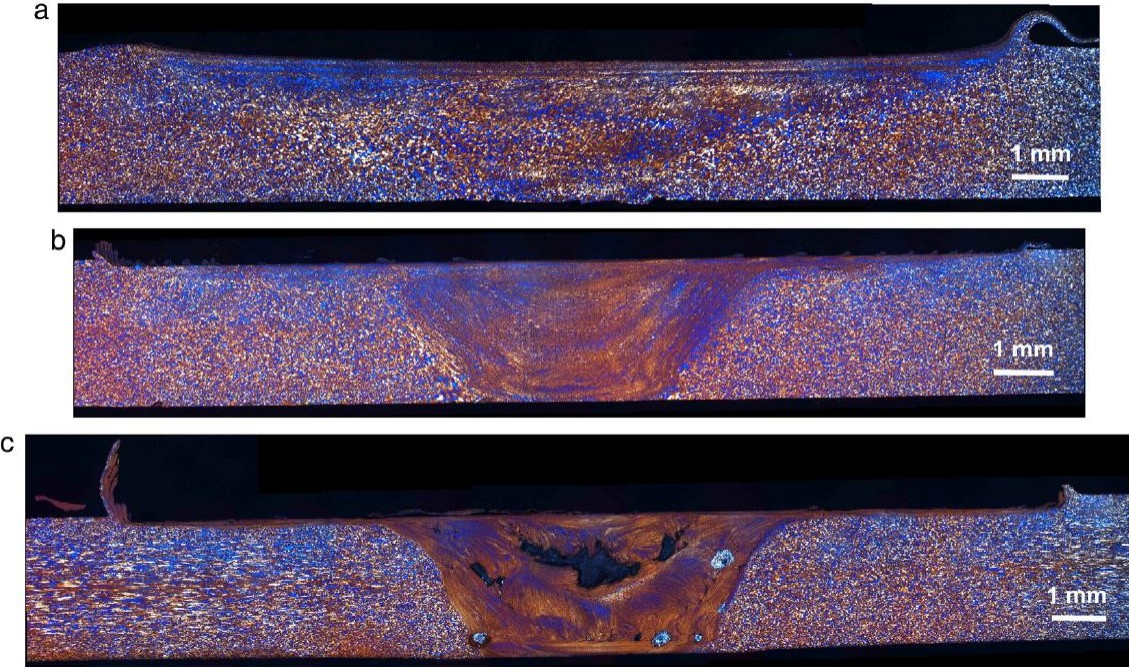

**Figure 11.** Montage of polarized optical microscopy (POM) of joints in as-received AA5754 alloy obtained by performing FSW at: (**a**) R = 0 mm, (**b**) R = 0.5 mm and (**c**) R = 1 mm.

On the basis of the microstructure and nanoindentation results herein reported, the soundness and improved mechanical and technological properties of the DS-FSW joints are inferred by the far more uniform hardness, $H$, and reduced Young's modulus, $Er$, values across the FSW joint. The observed better formability of the DS-FSW is believed to be microstructurally related, to one hand, to the nanoindentation hardness profile across FSW TMAZ and NZ, with respect to the base metal (BM), and, on the other hand, to a better Young's modulus response obtained across the FSW joint with respect to the BM. A high level of $H$ and $E_r$ uniformity was obtained using the pin–pin DS-FSW. This is thus believed to be the better compromise between a technological and a microstructural viewpoint.

As for the microstructural modification induced by the pin deviation from centreline in the non-age hardening AA5754, the R = 0 mm equiaxed grains characterize the whole extension of the FSW sheet. Under that experimental condition, the mean grain size throughout the welded zone (HAZ, TMAZ of the AS and RS) was essentially same as was observed at the BM. On the other hand, at R = 0.5 mm,

fine equiaxed recrystallized grains were induced to form throughout the HAZ, and TMAZ, in both the AS and the RS. Anyhow, in this case, the NZ was characterized by a diffuse presence of very coarse irregular grains; i.e., mixed fine recrystallized grains and coarser abnormal grain growth. The very coarse grains were flattened and pinched off in the central region of the top surface of the FSW section. This flattened grain region coincided with the shoulder diameter, also considering the entire pin deviation excursion of 2R = 1 mm. In the surrounding zones, coarse grains appeared due to the effect of the PWA (Figure 12). The grained structure obtained by R = 1 mm pin deviation (Figure 12c) appeared to be similar to the one induced by the R = 0.5 mm pin deviation (Figure 12b). Anyhow, in these FSW conditions, marked and coarse oxide zones were formed at the NZ. This important microstructural aspect makes the R = 1 mm pin deviation a useless technological condition. These microstructural comparisons made possible identifying the R = 0.5 mm pin deviation from the centreline as the most appropriate and promising FSW new approach for PWA non-age hardening aluminum alloys. Further details on the forging effect induced by the probe upon centreline deviation can be found in [16,17].

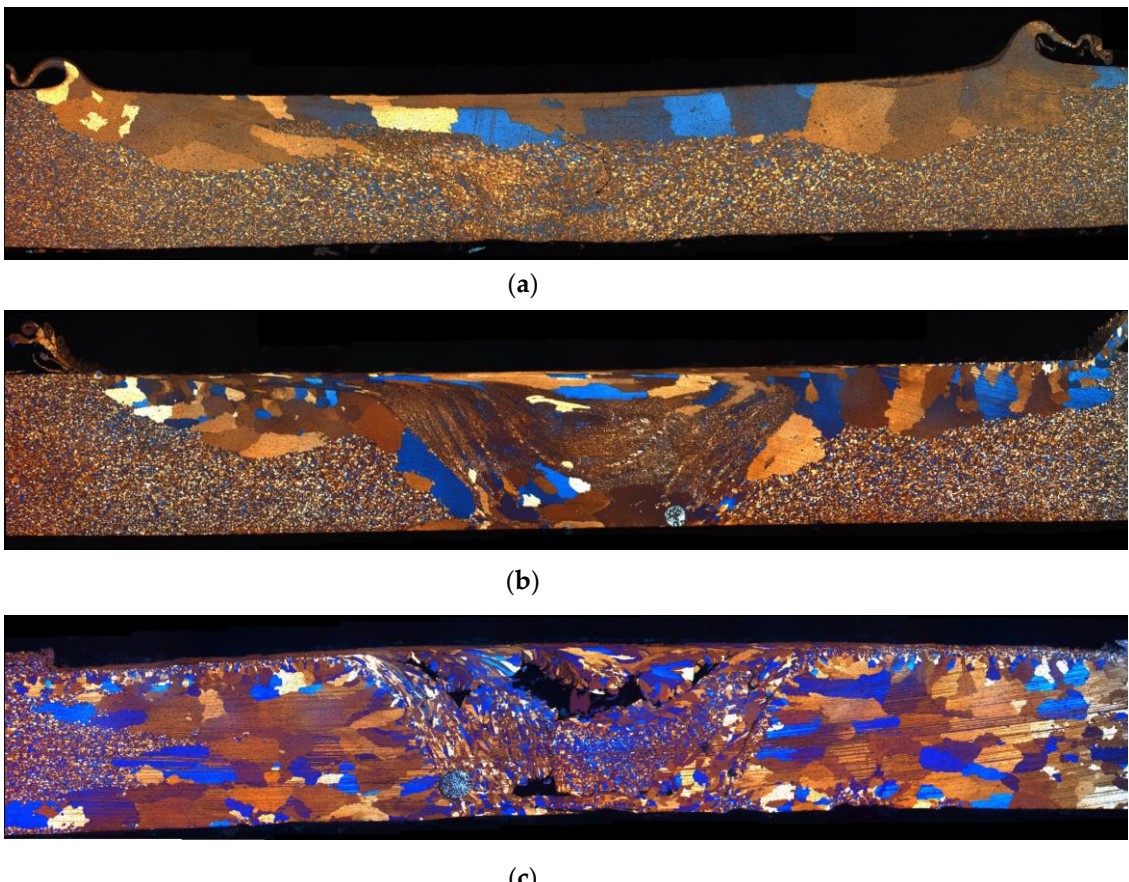

(**a**)

(**b**)

(**c**)

**Figure 12.** Montage of POM images of joints in AA5754 alloy obtained by performing FSW and post-welding annealing treatment (PWA at 415 °C/3 h, followed by furnace cooling). The effect of pin deviation from welding line on microstructure of joints: (**a**) R = 0 mm, (**b**) R = 0.5 mm and (**c**) R = 1 mm.

In conclusion, this paper reports the main results obtained by performing the two new approaches to the FSW. In particular, the authors highlighted that the DS-FSW technology is extremely useful for improving the post-welding formability of welded joints compared to the traditional FSW, both in terms of limiting dome height and forming limit curves. Such a result can be attributed to the higher uniformity of the hardness and Young's modulus across the sheet section, with respect to the FSW. Furthermore, based on the microstructure evidence, and the obtained mechanical response, the use of a RT-type FSW configuration can be encouraged only when FSW blanks are homogenized after FSW.

There is no need for deviating the pin tool from the welding line during FSW, as long as the AA5754 sheet is not annealed.

## 4. Conclusions

In this study two different evolutions of FSW were introduced and discussed. These were found to improve, significantly, the mechanical responses of the welded sheets compared to conventional FSW techniques. One FSW improvement consisted in the practice of a double side welding, and this was successfully applied to an age-hardenable AA6082 alloy. A second FSW innovative technique consisted of a small lateral deviation from the centreline transverse motion of the rotating pin during welding. This second approach was applied to a widely used, non-age-hardenable AA5754 sheet. Both cases were studied and characterized by subjecting the welding sheets to different meaningful heat treatments; that is, a T6-treatment for the AA6082, and full annealing for the AA5754. The resulting advantages of using these two new FSW approaches were accounted in terms of mechanical response and microstructural modifications across the welded NZ, TMAZ and HAZ.

The main results can be summarized as follows:

a.  Double side FSW (DS-FSW): (i) Elastic modulus, limiting dome height (LDH) and the local hardness showed a significant uniformity across the sheet section with respect to conventional FSW; (ii) homogeneous fine recrystallized (RX) grains were formed across the NZ, while in conventional FSW, coarser RX grains were observed; (iii) the microstructural and mechanical homogeneities generated a better formability (LDH) of the DS-FSW AA6082 sheet, compared to the conventional FSW. This interesting result held for pin and pinless welding processes.

b.  Pin deviation form centreline (RT): (i) The best RT FSW settings were found for a pin deviation from centreline by 1/30 (0.5 mm) of the shoulder diameter, using a truncated conical shaped pin with heights of 2.0 and 2.3 mm in a 2.5 thick AA5754 sheet. With these FSW set-ups, the post-welded annealed sheet showed the most significant mechanical improvements, among the different experimental conditions tested, over the conventional FSW technique; (ii) the induced microstructural modifications across the NZ, TMAZ and the HAZ were driven by geometric dynamic recrystallization (DRX) mechanism. This was ultimately induced by the combined effect of the shoulder pressure (heat input), and the post-welding annealing thermal energy.

c.  FSW emerged as an advanced technique for the joining of similar/dissimilar metals and alloys to be potentially adopted in various manufacturing and engineering application fields, such oil and gas pipelines, automobile part joining and aviation joining. The latest new approaches well addressed and accounted for FSW's widening potential to include new metal processing applications, wider light alloy candidates and new metallurgical production techniques, such as the additive manufacturing (AM). In this respect, the two herein presented new approaches showed a clear mechanical improvement of FSW joints in the case of aluminum alloy sheets, both non-age-hardening and age-hardening. New FSW approaches and methods, such the ones presented, make this joining technique one of the most prominent and promising welding techniques to foster and promote new technology frontiers in the making and repairing processing of light metal sheets and plates. Indeed, recent metallurgical developments, namely, additive manufacturing (AM), showed the applicability of the FSW to complex parts and components, promoting the adoption of this technique to a manufacturing scale. In this regard, the combination of friction stir processing (FSP) with other derived friction non-melting strengthening methods, can be nowadays considered as a direct evolution step of the FSW from which it originated, able to scale up all the possible application fields of this non-fusion metal attrition technique.

All the herein-mentioned major findings are believed to constitute a viable scientific basis for further technological improvements and challenges in FSW, and possibly all the nowadays-tested different friction stir processing routes, of light-alloys.

**Author Contributions:** M.C. and M.S. conceived and designed the experiments and prepared the manuscript; M.C. performed the OM inspections; A.F. contributed to conceiving the new approaches herein described and discussed the mechanical test results; S.S. analyzed the data, prepared the plots, and the revised manuscript; E.S. performed the mechanical tests and FSW; C.P. prepared the samples for the tests and performed the nanoindentation tests; M.S. performed the formability tests. All authors have read and agreed to the published version of the manuscript.

**Funding:** This research received no external funding.

**Conflicts of Interest:** The authors declare no conflict of interest.

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
