# Peer review of "New Approaches to Friction Stir Welding of Aluminum Light-Alloys"

_metals, doi:10.3390/met10020233_

Round 1

Reviewer 1 Report

The manuscript of Marcello Cabibbo and coauthor presents a new approaches to friction stir welding for light alloys. The manuscript idea is interesting, but the preset for must be substantially amended before publishing in the Metals journal.

Additional criticism are:

Page 3, line 103, I thing that the authors meant “HAZ” not “HAF”

Page 7, line 286: “Finally, nanoindentation was used” please mention the used equipment. This observation apply to all other used techniques. There is no mention about the used type of optical microscopy for example.

In abstract authors mention a SEM analysis, it is difficult to see in the manuscript where the images were recorded using SEM and optical microscopy.

Fig. 6 represent a computer drawing or is a photography of the actual process. Later should be better for the manuscript.

Authors apply the innovative techniques for too materials, but the data are mixed in the paper. The presented information should be consistent with both materials.

Reviewer 2 Report

Dear authors,

I find the manuscript quite difficult to read and understand. Most importantly, the actual goal of the work is not clear from the beginning, nor in the actual content. The paper lacks a clear structure. The English has also to be revised, but even more important, there are some terminologies related to FSW that have to be updated or used correctly. Please find enclosed the manuscript with some questions and suggestions. 

Further critical aspects:

1) Which is the contribution to the state of the art in FSW? This has to be stated more clearly. 

2) There is no Materials and Methods section (or Experimental).  Which equipment was used for welding and testing? How many replicates for each type of weld/testing? 

3) Most of the results need to be discussed and explained, also with literature references.

Good luck!

Reviewer 3 Report

Review work on FSW technology. The authors have reviewed the literature and described in general terms the impact of technological parameters of the FSW process on the structure and mechanical properties of light alloy sheets, mainly aluminum alloys. As an added value, they described the results of research on  new FSW technologies, i.e. DS-FSW and RT-FSW.
The test results should be supplemented with a prescriptive description of the mechanism of joint formation, e.g. through a detailed interpretation of the results of metallographic tests, results of microanalysis of the chemical composition of the disclosed components and their identification.
Figures 8 and 9 are illegible and not fully explained in the text.
The article can be published after completing the results of studies on the properties of joints and analysis of the impact of using new technologies on the mechanisms of joint formation and structure.

Round 2

Reviewer 1 Report

The authors have improved the manuscript and organized to be more readable and logic.

After analyzing the modifier manuscript, I believe that it deserves to be published in the Metals journal.

Author Response

We do thank you this Reviewer for having accepted our responses to the comments made to the manuscript.

Reviewer 2 Report

Thank you for the updates and answers. The understanding is now much improved. Nonetheless, there are still some issues to address. Please find enclosed the manuscript with further comments and suggestions. 

Good luck! 

Author Response

We have addressed allthe major and minor issues still pending. Thus, the uploaded 2nd revised version of our manuscript is a marked word file with all the needed changes, text modifications, and clarifications.

The authors believe that this can meet all the 2nd round revision requirements rised by this Reviewers. 

Please note, that both the two other Reviewers, and the Editor as well, had already considered the Rev 1 version suitable for publication.   

Reviewer 3 Report

The authors made corrections to the article, supplementing it with the deficiencies indicated. The article can be published in this form.

Author Response

(The authors gave the same response as above.)
